# Morphology and Selected Properties of Modified Potato Thermoplastic Starch

**DOI:** 10.3390/polym15071762

**Published:** 2023-04-01

**Authors:** Regina Jeziorska, Agnieszka Szadkowska, Maciej Studzinski, Michal Chmielarek, Ewa Spasowka

**Affiliations:** 1Łukasiewicz Network-Industrial Chemistry Institute, Rydygiera 8, 01-793 Warsaw, Poland; 2Department of High-Energetic Materials, Faculty of Chemistry, Warsaw University of Technology, Noakowskiego 3, 00-664 Warsaw, Poland

**Keywords:** starch, halloysite, hot water resistance

## Abstract

Potato thermoplastic starch (TPS) containing 1 wt.% of pure halloysite (HNT), glycerol-modified halloysite (G-HNT) or polyester plasticizer-modified halloysite (PP-HNT) was prepared by melt-extrusion. Halloysites were characterized by FTIR, SEM, TGA, and DSC. Interactions between TPS and halloysites were studied by FTIR, SEM, and DMTA. The Vicat softening temperature, tensile, and flexural properties were also determined. FTIR proved the interactions between halloysite and the organic compound as well as between starch, plasticizers and halloysites. Pure HNT had the best thermal stability, but PP-HNT showed better thermal stability than G-HNT. The addition of HNT and G-HNT improved the TPS’s thermal stability, as evidenced by significantly higher T_5%_. Modified TPS showed higher a Vicat softening point, suggesting better hot water resistance. Halloysite improved TPS stiffness due to higher storage modulus. However, TPS/PP-HNT had the lowest stiffness, and TPS/HNT the highest. Halloysite increased T_α_ and lowered T_β_ due to its simultaneous reinforcing and plasticizing effect. TPS/HNT showed an additional β-relaxation peak, suggesting the formation of a new crystalline phase. The mechanical properties of TPS were also improved in the presence of both pure and modified halloysites.

## 1. Introduction

As the materials of the millennium, plastics have become synonymous with technical and economic progress [1,2]. Unfortunately, the growing amount of polymer waste collected in landfills has a negative impact on the plants, animals, and people’s living conditions [3,4]. To reduce the negative effects on the environment, research is being conducted on the production of polymers derived from renewable and biodegradable raw materials [5,6,7]. Currently, the share of biodegradable plastics in the global plastics market is small, but shows an upward trend. In the modern world, caring about the natural environment is becoming not only an obligation, but also a necessity [8,9,10]. According to the EU Directive (SUP) on the ban on the production (from 2021) of certain single-use plastic products (cutlery, plates, stirrers, expanded polystyrene packaging), manufacturers of these products are obliged to replace them with biodegradable materials [4,6,11].

One of the most widespread natural polymers is starch [12,13,14]. Its biodegradability and bio renewability make this polymer “double green”, and it meets the principles of “green chemistry”. Great hopes for solving the problem of waste, especially packaging waste, are associated with the use of thermoplastic starch (TPS), which can be processed using traditional techniques such as extrusion, injection, pressing, etc. The advantages of starch are its low price and universal availability. The disadvantages, limiting its widespread use in industry, are the values of its glass transition and melting point (230−240 °C), which are higher than the thermal decomposition temperature (220 °C) [15].

Starch is produced from many different renewable sources. It is a cheap and inherently biodegradable material. It can be completely converted by microorganisms into carbon dioxide, water, minerals, and biomass, without negatively impacting the environment [16]. Starch grains come in all shapes and sizes, ranging from 2 µm to over 100 µm, depending on their botanical origin and type [17,18].

Starch contains mainly of 15–35 wt.% linear amylose and 65–85 wt.% non-linear amylopectin. The association of amylose and amylopectin, in the native state, gives a semicrystalline structure [2,13]. It is well known that native starch is brittle, and its mechanical properties are very poor. Lack of water causes thermal degradation to occur below the glass transition temperature [19]. Thanks to plasticizers, thermal processes, and shear stress, it is possible to convert native starch into a thermoplastic polymer [9,14,20]. However, due to poor mechanical properties and high hygroscopicity, plasticized starch cannot yet compete with polymers from non-renewable raw materials. To overcome these problems, nanocellulose [21], montmorillonite [22,23], kaolin [24], bentonite [25], nano silica [26], chitosan [27,28] and halloysite [29,30] have been used.

Halloysite, due to its structure consisting of hollow nanotubes and multiple rolled aluminosilicate layers, is physically and chemically similar to kaolinite [30,31,32]. The nanotubes’ outer diameter is 50–200 nm, their inner diameter is 5–30 nm, and their length is 0.5–25 µm. The negatively charged SiO_2_ forms the outer surface and the positively charged Al_2_O_3_ forms the inner surface of the HNT. Halloysite, thanks to high specific surface area, good thermal stability, low cost, and unique surface properties, is one of the most important clay minerals [33,34].

Schmitt et al. [35] found that the addition of 2–8 wt.% unmodified and modified (quaternary ammonium salt with benzalkonium chloride) halloysite nanotubes improved thermal stability and tensile properties of starch. The modified halloysite had a much higher Young’s modulus than the unmodified one.

Therefore, it can be expected that even a small addition of HNT will increase thermoplastic starch’s resistance to hot water and improve its mechanical properties. As far as we know, no comprehensive research on the use of pure halloysite and halloysite containing glycerol or polyester plasticizer for the modification of thermoplastic potato starch has been described yet. It was expected that HNT would act as a plasticizer and a reinforcing agent simultaneously.

Therefore, this article presents the effect of unmodified and modified with glycerol or polyester plasticizer halloysite on the thermal, mechanical, and morphological properties of thermoplastic potato starch. Before modification, pure HNT was subjected to ultra-sonication to introduce more organic compound into its porous structure and improve molecular interactions between TPS components. The structure and thermal properties were analyzed by Fourier infrared spectroscopy (FTIR), scanning electron microscopy (SEM), dynamic thermomechanical analysis (DMTA), thermogravimetric thermal analysis (TGA), differential scanning calorimetry (DSC) and Vicat softening temperature. Moreover, its tensile and flexural properties were determined.

## 2. Materials and Methods

### 2.1. Materials

Native potato starch with amylose content of 21%, pH 5.5–7.5 was obtained from B.E.S.T. Company, Parczew, Poland. Before processing, the starch was conditioned at 23 °C and 50% relative humidity for 48 h. Under these conditions, native starch contained 11% moisture. Glycerol (99% purity) (Sigma-Aldrich, Munich, Germany) and sorbitol (Sigma-Aldrich) were used as plasticizers. Pure (HNT) and modified with glycerol (G-HNT) or polyester plasticizer (PP-HNT) halloysite nanotubes were used as a plasticizer and reinforcer, simultaneously (Table 1). Pure halloysite was delivered from the open pit mine “Dunino’’, Intermark Company, Gliwice, Poland. The polyester plasticizer was obtained by the method described in [36]. Briefly, the plasticizer was obtained in a three-stage process. In the first stage, 2-butyl-2-ethyl-1,3-propanediol (BEPD), 2-methyl-1,3-propanediol (MPD), ethylene glycol (EG), and propylene glycol (PG) were placed into the reactor in appropriate proportions. After starting the stirrer and heating, a Fascat 4100 catalyst and succinic acid were added. The synthesis was carried out in an inert gas atmosphere at a temperature below 177 °C. After collecting about 81% of the theoretical condensate amount (water with a small glycols content), the process was continued under reduced pressure at a temperature of 175−180 °C until the acid number (AV) and hydroxyl number (HV) of 111.3 mg KOH/g and 1.7 mg KOH/g, respectively, were obtained. In the second step, Fascat 4100 catalyst and 2-ethylhexyl alcohol (EHA) were added. The process was carried out in an inert gas atmosphere at a temperature of 160−186 °C to obtain AV of 5.2 mg KOH/g. In the third stage, distillation of unreacted EHA was carried out at a temperature of 180−189 °C under reduced pressure. After obtaining an AV of 3.6 mg KOH/g and volatility of 0.38 wt.%, the process was continued for 3 h. A plasticizer with the following parameters was obtained: AV 2.2 mg KOH/g; HV 3.6 mg KOH/g; viscosity (25 °C) 2550 mPa·s; volatility (2 h/140 °C) 0.26 wt.%; M_w_ 2340 g/mol.

### 2.2. Halloysite Modification and Characterization

Modified halloysite was obtained in a two-stage process published elsewhere [37]. In the first stage, the suspension of native halloysite in demineralized water was subjected to an ultrasonic field with a power of 250–350 W and frequency of 20–40 kHz for 2–3 h. After ultra-sonication, a solution of modifying agent in ethyl alcohol was added to the suspension of halloysite in water and stirred for 3 h using a mechanical stirrer in an ultrasonic field. After evaporation of the alcohol, the precipitate was ground in a ball mill.

A BET-N2 sorption method (Tri Star II 3010, Micromeritics, Norcross, GE, USA) was used to determine the specific surface area of halloysite nanoparticles. The morphology of pure and modified halloysite nanotubes was studied by scanning electron microscopy (SEM) at 20 kV, using a Jeol JSM-6490LV microscope (Japan).

### 2.3. Thermoplastic Starch Processing

Plasticized starch was obtained in a two-stage process published elsewhere [37]. At first, 69 wt.% native starch, 10 wt.% glycerol and 20 wt.% sorbitol, and 1 wt.% of halloysite were blended in a laboratory mixer (LMX5-S-VS, Labtech Engineering Co. Ltd., Thailand) for 2 min at a speed of 1000 rpm. Then, the stirrer speed was reduced to 400 rpm and glycerol was introduced, after which the stirrer speed was increased to 1300 rpm. After reaching a temperature of 65 °C, the process was terminated. The homogenized starch was dried at 110 °C for 2 h (to a water content of about 10%), and then plasticized in a KraussMaffei Berstorff (Munich, Germany) twin-screw extruder with a screw diameter of 25 mm and a length of 51D. The process was carried out at a temperature of 40–160 °C and a screw speed of 100 rpm. Test samples were injection molded at 155–175 °C using an Arburg 420 M single screw injection machine (Allrounder 1000-250, Lossburg, Germany). The mold temperature was 20 °C.

### 2.4. Methods

Fourier infrared spectroscopy (FTIR) (Thermo Fisher Scientific, model Nicolet 6700, Waltham, MA, USA) was used to analyze the chemical structure. The spectra were recorded using at least 64 scans with 2 cm^−1^ resolution, in the spectral range of 4000–500 cm^−1^, using a KBr pellets technique. At least three samples of each halloysite and starch were evaluated.

The starch morphology was characterized by SEM in high vacuum using a Joel JSM 6100 microscope (Tokyo, Japan) operating at 20 kV. The impact fractured surface of the injection molded samples was coated with a thin gold layer to avoid charging and increase image contrast. At least three samples of each starch were evaluated.

The viscoelastic properties were studied by torsion dynamic thermomechanical analysis (DMTA) using a dynamic analyzer (Rheometrics RDS 2, Rheometric Scientific Inc., Piscataway, NJ, USA) at a frequency of 1 Hz. The strain level was 0.1%. Data were collected from −150 to 100 °C at a heating rate of 3 °C/min. The specimens (38 mm × 10 mm × 2 mm) were cut from injection molded samples. At least three samples of each starch were evaluated.

Thermal stability was determined by thermogravimetric analysis (TGA) using a TGA/SDTA 851e thermogravimetric analyzer (Metler Toledo, Greifensee, Switzerland) in a nitrogen atmosphere at heating rate of 10 °C/min, from 25 to 700 °C. The temperatures of 5% weight loss (T_5%_), 10% weight loss (T_10%_), and total weight loss at 700 °C were determined. In addition, the maximum decomposition rate temperature (T_max_) was determined from the differential thermogravimetric curve, which is the first derivative of the TG curve. The minima visible on DTG curve correspond to T_max_. At least three samples of each halloysite and starch were evaluated.

Differential scanning calorimetry (DSC) analysis was carried out using a DSC 822e apparatus (Metler Toledo, Switzerland) at a temperature from −50 °C to 200 °C in an argon atmosphere. The heating rate was 10 °C/min, and the argon flow rate was 50 mL/min. At least three samples of each halloysite and starch were tested.

The Vicat softening point was determined using HV3 apparatus by CEAST, Italy, according to ISO 306, method A (load 10 N, heating rate 50 °C/h).

The tensile and flexural properties were measured using an Instron 5500R universal testing machine (Massachusetts, UK) according to ISO 527 and ISO 178, respectively. The crosshead speeds for tensile and flexural tests were 5 and 2 mm/min, respectively. The gage length for tensile tests was 50 mm. The average of five measurements was taken as the result.

## 3. Results and Discussion

### 3.1. Halloysite Characterization

BET confirmed that before and after ultra-sonication, pure HNT had a porous structure with an average pore size of 11 nm and 17 nm, respectively (Table 1). At the same time, the average particle size increased from 93 nm to 173 nm. PP-HNT and G-HNT also showed a porous structure, and their average pore size gradually increased from 11 nm in HNT to 24 nm and 37 nm, respectively. A significant reduction in the specific surface area of the halloysite after modification and an increase in the size of pores and particles compared to pure halloysite indicated the build-up of the applied organic compound on the HNT surface (Table 1), confirmed by SEM, as shown in Figure 1. These observations are consistent with other studies [38,39].

#### 3.1.1. Infrared Spectroscopy (FTIR)

FTIR spectra of pure and modified halloysite are presented in Figure 2 and Figure 3. In the hydroxyl region (4000–3000 cm^−1^), the HNT spectrum shows two distinct peaks at 3690 and 3618 cm^−1^, corresponding to the stretching vibration of the Al_2_-OH groups’ inner-surface and the deformation vibration of interlayer water, respectively [31]. The weak peak at 1630 cm^−1^ is related to the OH stretching and deformation vibration of the adsorbed water, respectively. The bands at wavenumbers 1113 and 1023 cm^−1^ are assigned to the Si-O in-plane stretching vibration and perpendicular Si-O-Si stretching vibration, respectively. The band at wavenumber 907 cm^−1^ corresponds to the O-H deformation of the internal hydroxyl groups, and that at 787 cm^−1^ is due to a symmetrical Si-O stretching vibration. The bands at 744 and 687 cm^−1^ are attributed to the perpendicular Si-O stretching vibration. All these observations suggest the presence of more than one type of water in the halloysite structure, and are consistent with previous reports on halloysite [31,40,41,42,43,44].

In the case of PP-HNT and G-HNT, it was clear that the characteristic peaks of HNT were maintained. In the G-HNT spectrum, the presence of new peaks at the wavenumbers 3296 cm*^−^*^1^ (OH vibration), 2947 cm*^−^*^1^ (asymmetrical CH stretching vibration), 2875 cm*^−^*^1^ (symmetrical CH stretching vibration), and 1455 cm*^−^*^1^ (CH bending vibration), belonging to glycerol, prove the interaction of glycerol with HNT. In addition, these bands were slightly shifted compared to glycerol, and their intensity decreased significantly. The band of adsorbed water at 1630 cm*^−^*^1^ shifted to 1636 cm*^−^*^1^. The bands at 1113, 998, and 787cm^−1^ shifted slightly to 1110, 993 and 783 cm*^−^*^1^, respectively. Compared to the pure HNT, the spectrum of PP-HNT (Figure 3) shows new peaks at 2954 cm*^−^*^1^ (C−H stretching vibration), 2926 cm*^−^*^1^ (asymmetrical C−H stretching vibration), 2862 cm*^−^*^1^ (CH_2_ stretching vibration), and 1636 cm*^−^*^1^ (C=O stretching vibration), belonging to succinic acid, proving the interaction of polyester plasticizer with HNT. The intensity of the bands decreased significantly compared to polyester plasticizer. The band at 787 cm^−1^ shifted slightly to 791 cm*^−^*^1^. All these observations suggest the interactions between halloysite and the organic compounds (glycerol, polyester plasticizer).

#### 3.1.2. Thermal Properties

The thermal decomposition of glycerol, polyester plasticizer, and pure and modified halloysite is summarized in Table 2 and shown in Figure 4. The TG curve of pure HNT shows two phases of weight loss (Figure 4a,b). The mass loss from 25 °C to 150 °C is due to the removal of physically adsorbed water, while the mass loss in the temperature range of 150–403 °C is related to the removal of interlayer residual water. The mass loss above 403 °C is due to Al-OH and Si-OH groups dehydroxylation [45,46]. In contrast, PP-HNT and G-HNT show three weight loss phases with a gradual loss up to 262 °C and 164 °C, respectively, followed by two steeper weight losses (Figure 4a,b). The first mass loss up to about 100 °C is attributed to the adsorbed water loss, the second loss is due to the glycerol or polyester plasticizer loss (Figure 4c,d), and the third is due to the halloysite dehydroxylation. There was a significant decrease in the 5% weight loss temperature (T_5%_) compared to pure HNT results from the presence of an organic compound (glycerol, polyester plasticizer) in the halloysite structure, which degrades at a much lower temperature (Figure 4c,d, Table 2). The HNT modification caused a decrease in a maximum rate of decomposition temperature (T_max3_) compared to the T_max_ of pure HNT (Figure 4b).

The DSC curves of pure and modified HNT show phase change as a function of temperature (Figure 5). Pure HNT and G-HNT clearly show typical endothermic peaks at 106 °C and 114 °C, respectively, where absorbed water melts. In contrast, PP-HNT shows two endothermic peaks at 56 °C and 88 °C, which can also be attributed to the melting of absorbed water. These results are consistent with the FTIR observations and other studies [47,48]. 

### 3.2. Thermoplastic Starch

#### 3.2.1. Chemical Structure and Molecular Interactions

The chemical structure and molecular interactions of native starch, plasticizers and halloysite were studied by FTIR spectroscopy. Unmodified and modified TPS show similar FTIR absorption bands (Figure 6). The broad band between 3660 cm^−1^ and 2990 cm^−1^ is associated with complex vibrational stresses of free, inter-, and intramolecular bonds of hydroxyl groups [49,50]. The peak at 2936 cm^−1^ is related to C–H stretching (–CH_2_) of the anhydrous glucose ring [51], and the peak at 1646 cm^−1^ to water resulting from the starch hygroscopicity [49]. It is known that the water present in starch is strongly and directly bound to the starch molecules through the ion-dipole interaction, which is stronger than the normal water-water bond [51]. The peaks between 1149 cm^−1^ and 1012 cm^−1^ indicate the interactions between starch and plasticizers molecules [51]. The peak at 998 cm^−1^ is attributed to intramolecular hydrogen bonding of hydroxyl groups or the water plasticizing effect [50]. Moreover, the intensity of the peaks with maximum at 1012 cm^−1^ and 998 cm^−1^ is lower for TPS/HNT and TPS/G-HNT compared to TPS, but higher for TPS/PP-HNT, demonstrating the molecular interactions between starch, plasticizers (glycerol/sorbitol) and halloysite. Similar observations on the interactions between starch and plasticizers have been reported in other studies [52,53].

#### 3.2.2. Morphology

The morphology of native and plasticized starch was observed by SEM, as illustrated in Figure 7. TPS shows a uniform morphology with no visible remains of starch granules, which suggests that the plasticizers (glycerol, sorbitol), together with heat and shear stress in the extrusion process, effectively destroyed the starch granules. SEM images indicate that booth pure and modified HNTs are evenly distributed in TPS matrix (Figure 7b–e). TPS, TPS/PP-HNT and TPS/G-HNT exhibit rough fracture surfaces (Figure 7b,d,e). However, the roughest surface is observed for TPS/PP-HNT (Figure 7d), while TPS/HNT shows the smoothest surface (Figure 7c). It is worth noticing to note that the addition of HNT and G-HNT improves TPS homogeneity, confirming the greater plasticizing effect. Based on the SEM results, it can be concluded that TPS/PP-HNT is the most flexible, while TPS/HNT the most brittle, which is also confirmed by the mechanical properties. 

#### 3.2.3. Viscoelastic Behavior

Dynamic thermomechanical analysis (DMA) was performed to investigate the interfacial interactions between starch and halloysite. Two distinct decreases in the storage modulus (G′) can be observed in Figure 8a. The first one at −50 °C corresponds to the β relaxation related to the crystalline structure of thermoplastic starch. The second decrease occurs in the temperature range from 30 to 70 °C and is associated with the α relaxation transition (T_α_), which probably corresponds to the melting process of thermoplastic starch. Compared to TPS, the storage modulus of TPS modified with halloysite shows a significant increase in the following order: TPS/G-HNT < TPS/PP-HNT < TPS/HNT, which indicates higher stiffness. This is due to the limitations of the starch chains’ segmental movement [9,54] and the presence of nanotubes with a high modulus and high aspect ratio [55]. This phenomenon is particularly evident above the glass transition of the starch-rich phase (T_α_).

In the case of TPS, the loss modulus (G″) shows two peaks at −21 °C and 50 °C (Figure 8b). The G″ peaks are known as polymer relaxations related to glass transition temperature (T_g_) or secondary transformations. The upper transition is attributed to the glass transition of the starch-rich phase (T_α_), while the lower one is attributed to the glass transition of the glycerol-rich phase (T_β_) [9,26,55]. An additional, a β relaxation peak is observed only for pure HNT, which suggests the formation of a new crystalline phase. All modified TPS show lower T_β_ and higher T_α_ (Table 3). A decrease in T_β_ suggests the presence of more flexible regions or greater mobility of the starch chains fragments in the crystalline phase, indicating a greater plasticizing effect. In contrast, an increase in T_α_ suggests the presence of a stiffer regions or less mobility of the starch chain fragments in the amorphous phase, confirming a bigger reinforcing effect. It can be concluded that HNTs behave simultaneously as a plasticizing and reinforcing agent. A similar phenomenon has been reported in other studies [9,26].

#### 3.2.4. Thermal Properties

Thermogravimetric analysis (TGA) was used to determine the temperature above which the degradation and destruction of thermoplastic starch begins. The addition of 1 wt.% halloysite affects the thermal stability of TPS, depending on its modification (Figure 9a, Table 4). The highest thermal stability was obtained for TPS/HNT, as evidenced by an increase in T_5%_ and T_10%_ by 19 °C and 17 °C, respectively, compared to unmodified TPS. The incorporation of G-HNT into TPS also increases the decomposition onset temperature (9 °C) and 10% mass loss temperature (3 °C). Thus, due to high thermal stability and good interaction with native starch, glycerol and sorbitol, HNT and G-HNT improve the thermal stability of starch [30,35]. However, as expected, PP-HNT slightly decreases the thermal resistance of TPS, as evidenced by lower thermal decomposition temperatures. These results are consistent with morphology analysis (Figure 7) and mechanical properties (Figure 10).

The DSC curves of unmodified and halloysite-modified TPS show no phase change as a function of temperature (Figure 9b), suggesting that TPS is fully amorphous. In general, halloysite slightly reduced the glass transition temperature (T_g_) of TPS. T_g_ was observed from −1.2 °C to 4.3 °C, depending on halloysite modification. TPS/PP-HNT showed the lowest T_g_, suggesting better flexibility, which was confirmed by mechanical properties (Figure 10) and SEM observations (Figure 7). 

An important parameter indicating resistance to hot water is the Vicat softening point. The addition of halloysite causes a significant increase in the Vicat temperature by 3–11 °C, suggesting better hot water resistance. The highest Vicat point was obtained for starch containing pure HNT, and the lowest for starch with the addition of G-HNT.

#### 3.2.5. Mechanical Properties

The tensile and flexural properties of unmodified and halloysite-modified TPS are presented in Figure 10. It is clear from Figure 10a that HNT reduces the elongation at break while increasing the tensile strength. In contrast, modified halloysite improves elongation. The highest elongation and tensile strength were obtained using PP-HNT, which confirms the plasticizing and reinforcing effect of halloysite. Moreover, TPS had significantly lower flexural strength and Young’s modulus than the modified with halloysite. In the case of TPS/HNT, the tensile strength and Young’s modulus increased up to 39% and 45%, respectively. After adding the modified halloysite, the tensile and flexural modulus of TPS decreased while the flexural strength increased, suggesting that the starch became more flexible compared to TPS/HNT. This explains the reduction in starch–starch intermolecular interactions, which results in an increase in the free volume and mobility of the starch chains. These findings are consistent with previous morphology analysis and DMTA results. The improvement of tensile and flexural properties results from the high aspect ratio and high mechanical properties of halloysites in combination with good interactions between starch, plasticizers and halloysite. Indeed, the good interfacial adhesion between the halloysites and the starch together with the uniform dispersion of the halloysites in the starch matrix enables effective stress transfer from the matrix to the reinforcement, resulting in an increase in tensile and flexural strength. Similar results were reported in other studies [35,52].

## 4. Conclusions

Potato thermoplastic starch containing pure halloysite, glycerol-modified halloysite or polyester plasticizer-modified halloysite was obtained in a twin-screw extruder. Ultra-sonication was used to introduce more organic compound into the porous structure of halloysite and improve molecular interactions between native starch, plasticizers and halloysite. BET, FTIR and SEM confirmed successive incorporation of glycerol and polyester plasticizer into halloysite. However, modified halloysite showed lower thermal stability then pure HNT due to the presence of the organic compound (glycerol, polyester plasticizer) in the halloysite structure, which degrades at a much lower temperature. FTIR confirmed the interactions between starch, plasticizers (glycerol, sorbitol) and halloysite. HNT and G-HNT, due to high thermal stability and good interaction with TPS components, increased its heat resistance. Moreover, HNTs improved stiffness of thermoplastic potato starch, as evidenced by the higher storage modulus, depending on the halloysite modification. TPS/HNT exhibited the highest stiffness, while TPS/PP-HNT the lowest stiffness. DMTA confirmed simultaneous reinforcing and plasticizing effect of halloysite, as evidenced by an increase in T_α_ and a decrease in T_β_. The mechanical properties were significantly improved after addition of pure and modified halloysites into starch matrix due to uniform distribution of halloysite in the starch matrix. Pure HNT showed a significantly higher Young’s modulus and flexural modulus than modified halloysites, but a lower elongation at break, tensile and flexural strength compared to PP-HNT. In addition, hot water resistance was improved due to a significantly higher Vicat softening point.

Potato starch is a cheap, renewable, easily available, and particularly good raw material for both physical and chemical modification, while halloysite is a natural and biocompatible raw material. Therefore, biodegradable starch-based materials modified with halloysite can replace conventional non-degradable polymers in single-use plastic products. Such studies are currently being conducted by our research group and will be the subject of a future publications.

## 5. Patents

The results of the work are protected by the Polish patent application P 441 782 (2022).

## Figures and Tables

**Figure 1 polymers-15-01762-f001:**
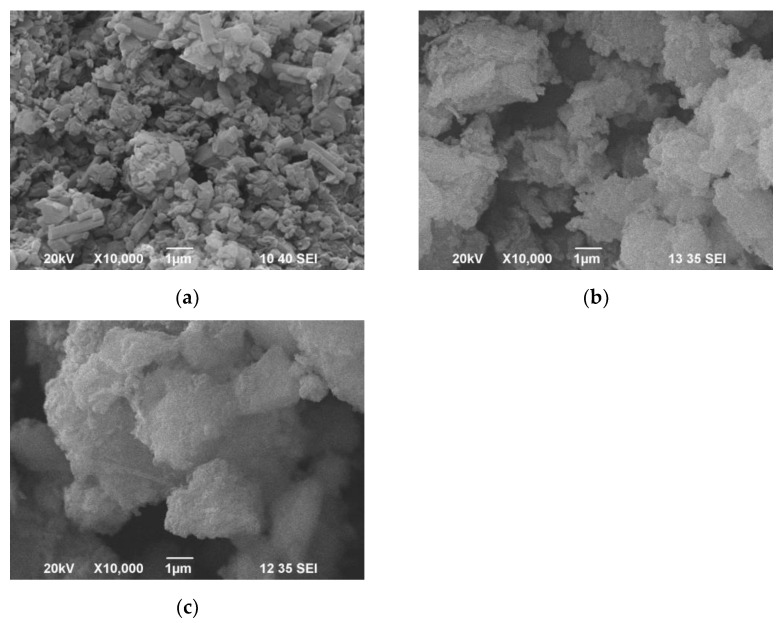
SEM images of pure HNT (**a**), PP-HNT (**b**) and G-HNT (**c**).

**Figure 2 polymers-15-01762-f002:**
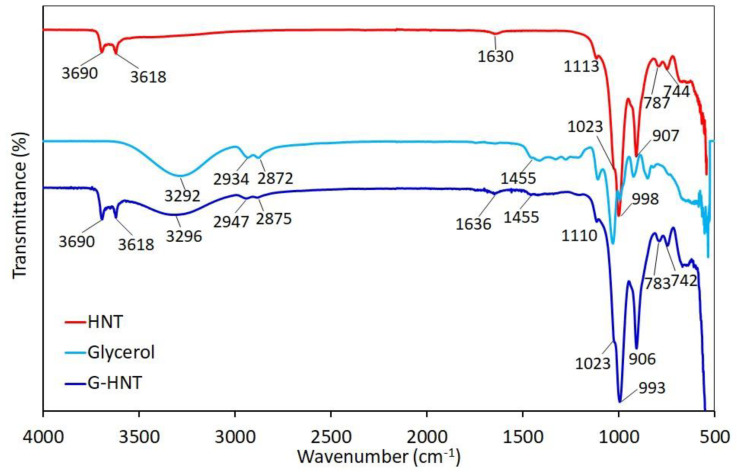
FTIR spectra of HNT, glycerol and G-HNT in the 4000–500 cm^−1^ region.

**Figure 3 polymers-15-01762-f003:**
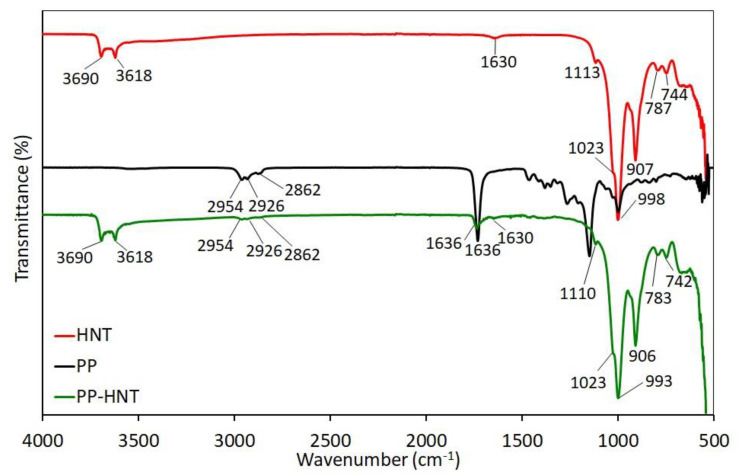
FTIR spectra of HNT, polyester plasticizer and PP-HNT in the 4000–500 cm^−1^ region.

**Figure 4 polymers-15-01762-f004:**
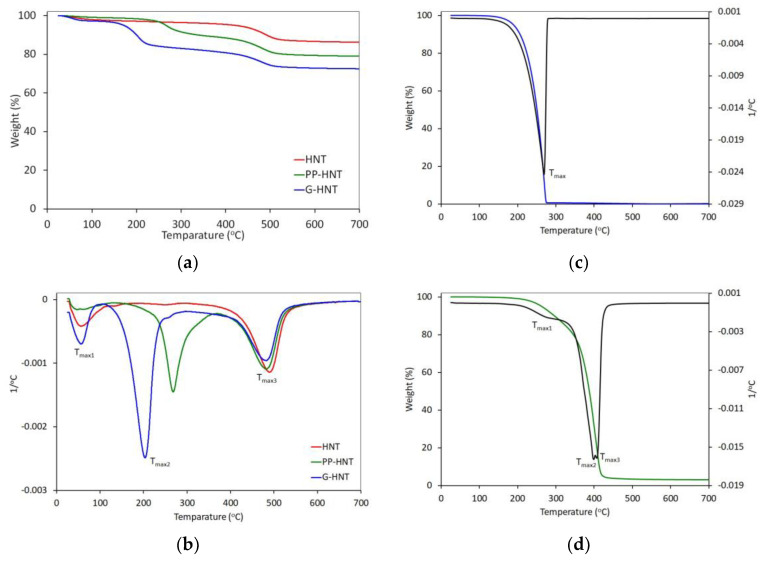
TG and DTG curves of halloysites (**a**,**b**), glycerol (**c**), and polyester plasticizer (**d**).

**Figure 5 polymers-15-01762-f005:**
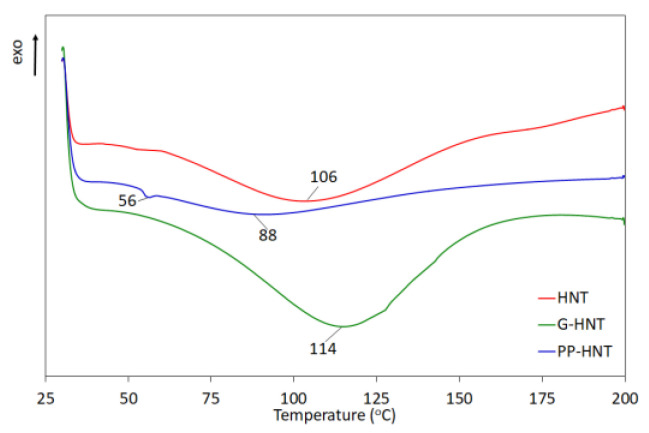
DSC curves of pure and modified HNT.

**Figure 6 polymers-15-01762-f006:**
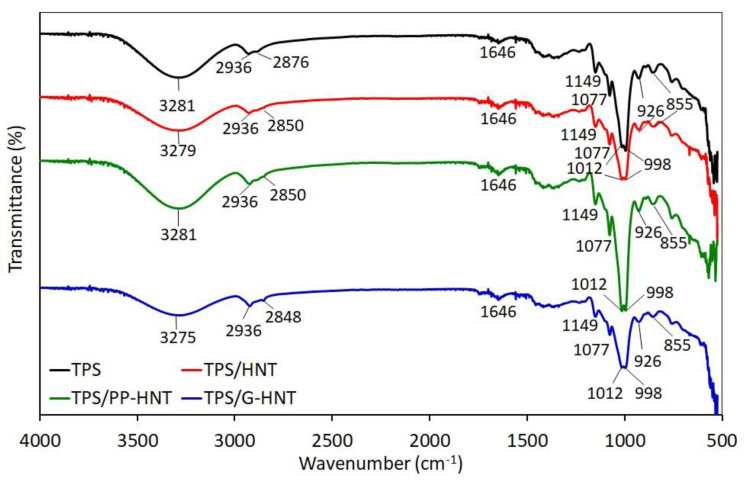
FTIR spectra of unmodified and halloysite-modified TPS in the 4000–500 cm*^−^*^1^ region.

**Figure 7 polymers-15-01762-f007:**
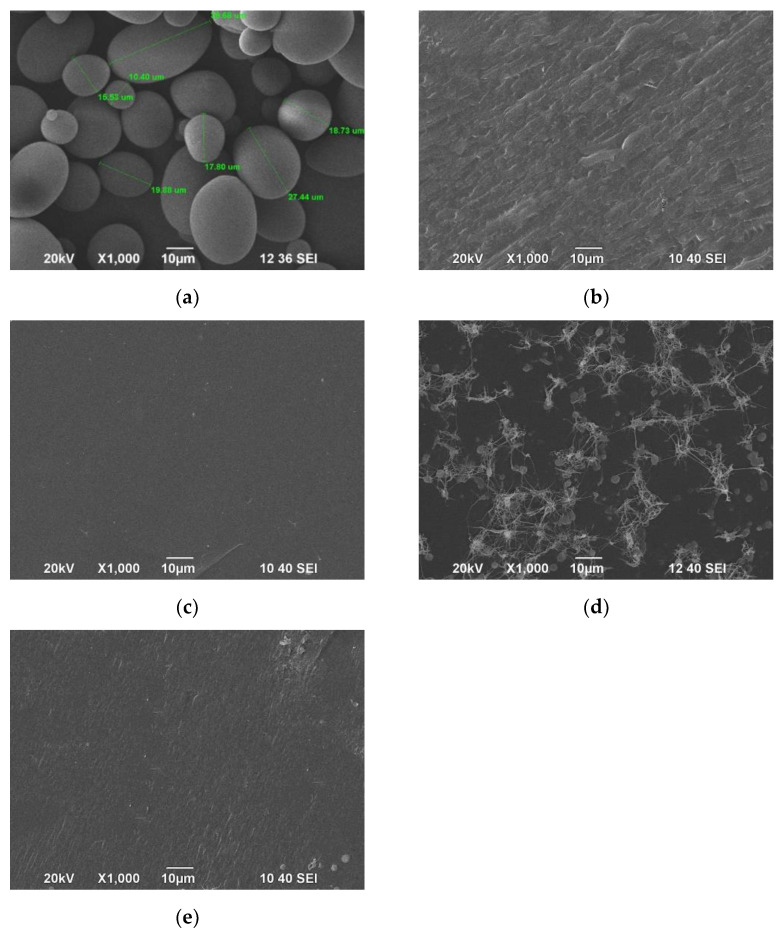
SEM images of native starch (**a**), TPS (**b**), TPS/HNT (**c**), TPS/PP-HNT (**d**) and TPS/G-HNT (**e**).

**Figure 8 polymers-15-01762-f008:**
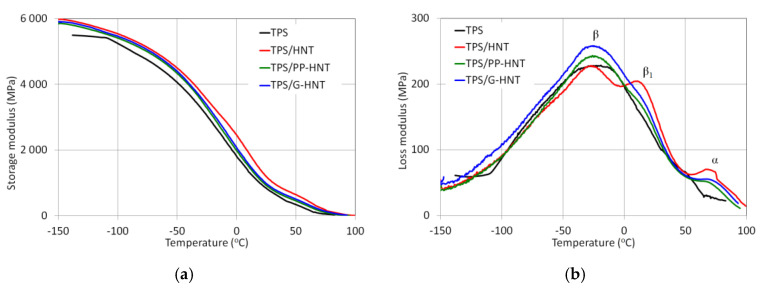
Storage modulus (**a**) and loss modulus (**b**) of unmodified and halloysite-modified TPS.

**Figure 9 polymers-15-01762-f009:**
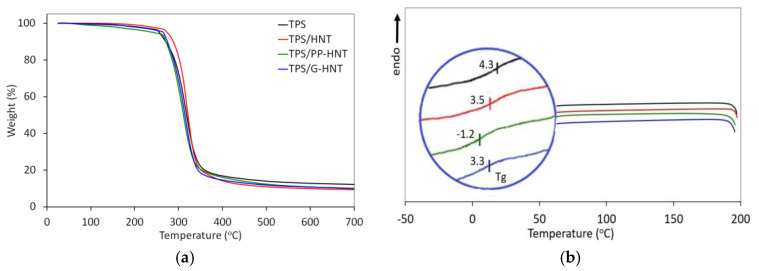
TGA (**a**) and DSC (**b**) thermograms of unmodified and modified TPS.

**Figure 10 polymers-15-01762-f010:**
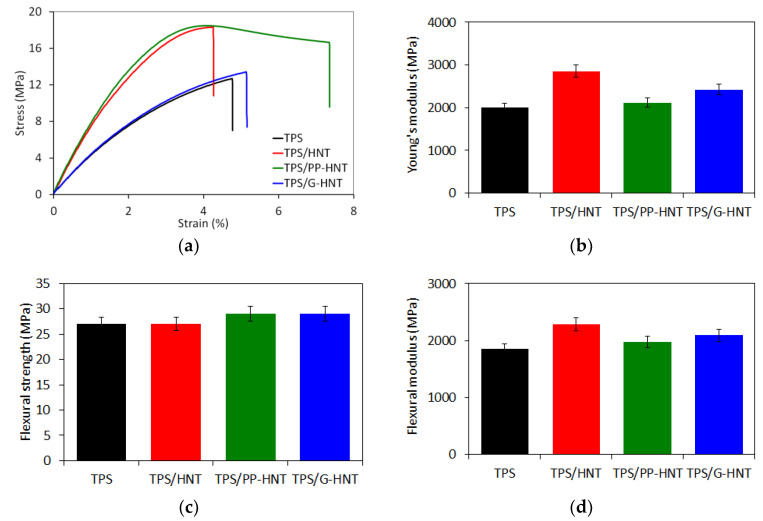
Stress–strain curves (**a**), Young’s modulus (**b**), flexural strength (**c**) and flexural modulus (**d**) of unmodified and halloysite-modified TPS.

**Table 1 polymers-15-01762-t001:** Pure and modified halloysite characteristic.

Halloysite	BET Surface Area (m^2^/g)	Average Pore Size (nm)	Average Particle Size (nm)
Pure HNT	61	11	93
HNT after ultra-sonication	35	17	173
PP-HNT	15	24	414
G-HNT	8	37	774

**Table 2 polymers-15-01762-t002:** TGA data of glycerol, polyester plasticizer, pure and modified halloysite.

Halloysite	T_5%_ (°C)	T_max_ (°C)	Residue (%)
Glycerol	182	258	0
Polyester plasticizer	261	275, 395, 407	3
Pure HNT	403	58, 491	86
PP-HNT	262	56, 268, 483	79
G-HNT	164	58, 203, 482	72

**Table 3 polymers-15-01762-t003:** Storage modulus and relaxation temperature of unmodified and halloysite-modified TPS.

Sample	Storage Modulus (MPa)		Relaxation Temperature (°C)		
	−21 °C	23 °C	T_α_	T_β_	T_β1_
TPS	2820	900	50	−22	–
TPS/HNT	3400	1380	70	−27	12
TPS/PP-HNT	3100	1040	69	−25	–
TPS/G-HNT	3090	1120	72	−25	–

**Table 4 polymers-15-01762-t004:** Thermal properties of unmodified and modified TPS.

Sample	T_5%_ (°C)	T_10%_ (°C)	Residue (%)	T_max_ (°C)	T_g_ (°C)	Vicat (°C)
TPS	253	268	12	313	4.3	65
TPS/HNT	272	285	10	315	3.5	76
TPS/PP-HNT	243	268	10	307	–1.2	72
TPS/G-HNT	262	271	10	313	3.3	67

## Data Availability

The data that supported the findings of this study are available from the corresponding author upon request.

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
