# Peer review of "Morphology and Selected Properties of Modified Potato Thermoplastic Starch"

_polymers, 2023, doi:10.3390/polym15071762_

Round 1

Reviewer 1 Report

This study prepared potato thermoplastic starch containing pure halloysite (HNT), glycerol-modified halloysite (G-HNT) or polyester plasticizer-modified halloysite (PP-HNT) by melt-extrusion, and their morphology, thermal stability, and mechanical strength, etc. has been characterized. The topic of this work is interesting for biodegradable plastic. Thus I’d suggest the acceptance of this work by minor reversion.

1. In Figure 10, can the authors show the sample’s stress-strain curves?

2. Figure 4 (c) is not ©.

3. Figure 9, the font in abscissa and ordinate are suggested to keep the same.

Author Response

Comments and Suggestions for Authors

  1. In Figure 10, can the authors show the sample’s stress-strain curves?

The article has been supplemented with stress-strain curves

  1. Figure 4 (c) is not ©.

It was changed.

  1. Figure 9, the font in abscissa and ordinate are suggested to keep the same.

It was changed.

Reviewer 2 Report

The manuscripts study morphology and selected properties of modified potato thermoplastic starch.

General comments

Abstract

Often the abbreviates did not use properly.

Keywords

Remove repetitive words that are similar to the title.

Introduction

The dynamic of writing descends from the fifth paragraph. The author must amend and clearly justify why halloysite is needed in solving the outlined problems.  

Results and discussion

3.1

Kindly plot the log distribution nominal for the pore size of each sample.

3.1.1

Figures 2 and 3 – Replot by increasing the peak intensity with the y-axis amendment scale.

3.1.2

Figure 4 – amend label “(c)”. Caption too.

Figure 4 – x-axis should use a similar scale. Change the colour of the second y-axis for DTG in (c) and (d)?

“…25°C to 403°C is due to the removal of physically adsorbed water and interlayer residual water.” – The temperature range is incorrect. The other components were degraded 100° to 403°C. Kindly clarify and justify.

Why Tmax has three values? Tmax should have one value only.

What is the significance of the TGA analysis of your study? What thermal analysis contributes to the end user?

3.2.1

Figure 6 - Replot by increasing the peak intensity with the y-axis amendment scale.

3.2.3

Why is the residue not as high as reported in Figure 4(a)

Table 3 – How you identified Tmax?

Conclusion

Conclude, not summarize. The conclusion should answer your objective(s).

Author Response

Abstract

Often the abbreviates did not use properly.

This has been verified in the article

 Keywords

Remove repetitive words that are similar to the title.

Keywords have been verified

 Introduction

The dynamic of writing descends from the fifth paragraph. The author must amend and clearly justify why halloysite is needed in solving the outlined problems.

It was supplemented in the article 

 Results and discussion

3.1

Kindly plot the log distribution nominal for the pore size of each sample.

We believe that the addition of four drawings that do not bring new information to the article is not justified. We leave the decision to the editors.

3.1.1

Figures 2 and 3 – Replot by increasing the peak intensity with the y-axis amendment scale.

Figures 2 and 3 were corrected

3.1.2

Figure 4 – amend label “(c)”. Caption too.

It was changed.

Figure 4 – x-axis should use a similar scale. Change the colour of the second y-axis for DTG in (c) and (d)?

The color has been changed.

“…25°C to 403°C is due to the removal of physically adsorbed water and interlayer residual water.” – The temperature range is incorrect. The other components were degraded 100° to 403°C. Kindly clarify and justify.

Why Tmax has three values? Tmax should have one value only.

Three Tmax are due to the composition of the material

What is the significance of the TGA analysis of your study? What thermal analysis contributes to the end user?

TGA analysis was used to determine the temperature above which degradation and destruction of thermoplastic starch begins.

3.2.1

Figure 6 - Replot by increasing the peak intensity with the y-axis amendment scale.

Figure 6 was corrected

3.2.3

Why is the residue not as high as reported in Figure 4(a)

Figure 4a was corrected

Table 3 – How you identified Tmax?

 The method of determining Tmax is included in the description of the TGA method

Conclusion

Conclude, not summarize. The conclusion should answer your objective(s).

The conclusions were verified as suggested by the reviewer

Reviewer 3 Report

The manuscript entitled “Morphology and Selected Properties of Modified Potato Thermoplastic Starch” by Jeziorska et al. investigates the effect of unmodified and modified with glycerol or polyester plasticizer halloysite on the thermal, mechanical, and morphological properties of thermoplastic potato starch. In general, it is an interesting and well designed research. I propose its publication after the revision of following comments. So, please:

1) Abstract: avoid the details e.g. “Before modification HNT was subjected to ultra-sonication” and present clearly the main conclusion of your research.

2) Line 136: give more details on the injection molding of the test samples.

3) 2.4 Methods: Give the number of repetitions in DMTA, TGA and DSC experiments and the standard deviation in the corresponding tables.

4) Lines 228-230: From Figure 4b it seems that pure HNT presents higher maximum rate of decomposition temperature Tmax3 compared to PP-HNT and G-HNT.

5)  Line 268: Figure 7c-e instead of Figure 7b-e

6) Correct the numeration of Tables.

7) Refer if any significant change in the residue of TPS was observed in the TGA experiments.

8) The Conclusion section is a repetition of the abstract. Try to point out the main findings of the research.

Author Response

1) Abstract: avoid the details e.g. “Before modification HNT was subjected to ultra-sonication” and present clearly the main conclusion of your research.

The abstract was verified as suggested by the reviewer. Detailed information is provided in the introduction.

2) Line 136: give more details on the injection molding of the test samples.

The article has been supplemented with more details on the preparation of test samples by injection molding

3) 2.4 Methods: Give the number of repetitions in DMTA, TGA and DSC experiments and the standard deviation in the corresponding tables.

This information was included in the description of research methods

4) Lines 228-230: From Figure 4b it seems that pure HNT presents higher maximum rate of decomposition temperature Tmax3 compared to PP-HNT and G-HNT.

It was corrected in the paper

5)  Line 268: Figure 7c-e instead of Figure 7b-e

This has been clarified in the article

6) Correct the numeration of Tables.

Numeration of tables was corrected

7) Refer if any significant change in the residue of TPS was observed in the TGA experiments.

The article was supplemented with the change of the TPS residue

8) The Conclusion section is a repetition of the abstract. Try to point out the main findings of the research.

The conclusions were verified as suggested by the reviewer